analytical chemistry

rupatadine, montelukast, native fluorescence, derivative synchronous fluorescence spectroscopy, pharmaceutical dosage forms

**Author for correspondence:**
Mohamed I. El-Awady
e-mail: mohamedelawady2@yahoo.com

This article has been edited by the Royal Society of Chemistry, including the commissioning, peer review process and editorial aspects up to the point of acceptance.

# Green quantitative spectrofluorometric analysis of rupatadine and montelukast at nanogram scale using direct and synchronous techniques

Rana Ghonim[1,2], Mohamed I. El-Awady[1,2],
Manar M. Tolba[1] and Fawzia Ibrahim[1]

[1]Department of Pharmaceutical Analytical Chemistry, Faculty of Pharmacy, Mansoura University, Mansoura 35516, Egypt
[2]Department of Pharmaceutical Chemistry, Faculty of Pharmacy, Delta University for Science and Technology, International Coastal Road, Gamasa 11152, Egypt

(iD) MIE-A, 0000-0003-2675-7530

Two green-sensitive spectrofluorometric methods were investigated for assay of rupatadine (RUP) [method I] and its binary mixture with montelukast (MKT) [method II]. Method I depends on measuring native fluorescence of RUP in the presence of 0.10 M $H_2SO_4$ and 0.10%w/v sodium dodecyl sulfate at 455 nm after excitation at 277 nm. The range of the first method was 0.20–2.00 µg ml$^{-1}$ with detection and quantitation limits of 59.00 and 179.00 ng ml$^{-1}$, respectively. Method II depends on the first derivative synchronous spectrofluorometry. The derivative intensities were measured for the two drugs in an aqueous solution containing McIlvaine's buffer pH 2.60 at fixed $\Delta\lambda$ of 140 nm. Each drug was estimated at zero-contribution of the other. The intensity was measured at 261 and 371 nm for RUP and MKT, respectively. The method was linear over 0.10–4.00 and 0.20–1.60 µg ml$^{-1}$ with limits of detection 31.00 and 66.00 ng ml$^{-1}$ and limits of quantitation 94.00 and 200.00 ng ml$^{-1}$ for RUP and MKT, respectively. The method was extended to determine this mixture in laboratory-prepared mixtures and combined tablets. Method validation was performed according to ICH guidelines. Statistical interpretation of data revealed good agreement with the comparison method. Method greenness was confirmed by applying three different assessment tools.

# 1. Introduction

Rupatadine (RUP) fumarate (figure 1*a*) is 8-Chloro-11-[1-[(5-methylpyridin-3-yl)methyl]piperidin-4-ylidine]-6,11-dihydro-5H-benzo[5,6]cyclohepta[1,2-b]pyridine(2E)-but-2-enedioate. Montelukast (MKT) sodium (figure 1*b*) is sodium[1-[[[1R)-1-[3-(E)-2-(7-choroquinolin-2-yl)ethenyl]phenyl]-3-[2-(1-hydroxy)methylethyl)phenyl]propyl]sulfanyl]methyl]cyclopropyl] acetate [1]. RUP and MKT are official drugs in British Pharmacopeia (B.P) [1] and United States Pharmacopeia [2]. RUP is a second-generation non-sedating antihistamine with platelet-activating factor antagonist activity that is prescribed for the treatment of allergic rhinitis, conjunctivitis and chronic idiopathic urticaria. It is given as the fumarate although doses are expressed in terms of the base; RUP fumarate 12.80 mg is equivalent to about 10.00 mg of RUP. The usual oral dose is the equivalent of 10.00 mg once daily of RUP. MKT is a selective leukotriene receptor antagonist that is approved in cases of allergic rhinitis and chronic asthma. It is also used as a prophylactic agent for exercise-induced asthma [3]. It is always co-administered with corticosteroids. RUP and MKT are co-formulated in tablet dosage forms like (Rupanex M®, Montyrup®) by pharmaceutical ratio 1 : 1. It has also been found that this combination is more effective than a single one in control of allergic rhinitis symptoms.

It is commonly known that spectrofluorometric methods are sensitive, selective, economic, accurate, rapid and usually green. However, some selectivity problems appeared, especially in multi-drug analysis due to the overlapping of their excitation and emission spectra as RUP and MKT. Synchronous fluorescence spectroscopy (SFS) solved this problem by providing simple, sharp spectra with high selectivity and low interference. Our method uses a constant difference between wavelengths, and it is a type of SFS known as constant wavelength SFS. So, SFS has an important feature over the conventional fluorescence which in turn improves the spectral resolving and diverging of light. Coupling the SFS technique with derivative amplitude leads to perfect resolution for both drugs [4].

The literature pointed to some reports for RUP estimation like densitometric [5], derivative UV spectrophotometry [6,7], RP-HPLC [8–11], GC-MS [12,13] voltametric techniques [14], RP-UPLC [15], LC-MS/MC [16], capillary zone electrophoresis [17], HPTLC [18], non-aqueous potentiometric titration method [19] and spectrofluorometric derivatization technique [20]. Different analytical methods have been used in determination of MKT such as RP-HPLC [21,22], UV spectrophotometry [23], spectrofluorometric [24], derivative spectrofluorometry [25] and voltametric technique [26]. MKT and RUP were estimated together by RP-UPLC and RP-HPLC methods with UV spectrophotometric detection [10,15]. It is worth mentioning that there is no spectrofluorometric method for determination of RUP alone (conventional) without reactions or with MKT (derivative synchronous fluorometric technique) in combination pharmaceutical dosage forms.

In the current study, we aim to determine RUP alone and in the presence of MKT as their co-formulation in a tablet dosage form. By scanning their native fluorescence spectra, great overlapping between them is so challenging. So, first derivative SFS (FDSFS) is a magnificent way to separate such mixtures qualitatively and quantitatively (method II) while the conventional native fluorescence technique is used for the determination of RUP (method I). These two sensitive spectrofluorometric methods are simple and highly green for the quantification of RUP and MKT in the commercial dosage forms.

# 2. Experimental

## 2.1. Apparatus

Cary Eclipse Fluorescence Spectrophotometer equipped with Xenon flash lamp. High-sensitivity mode (800 V), smoothing factor 20.00, slit width 5.00 nm with 1.00 cm quartz cell was manipulated for the conventional spectrofluorometric measuring method.

Synchronous spectrofluorometric measurements were performed at $\Delta\lambda = 140$ nm with scanning in the range of 200–600 nm. Gathering the stored data was achieved by the Cary Eclipse software. The first derivative spectra were manipulated at a filter size of 19.00 and an interval of 1.00 nm. A scan rate of 600 nm min$^{-1}$ was adopted using 10 nm excitation and emission windows. A pH-meter (Consort, NV P-901, Belgium) was used for adjusting the pH of buffer solutions. A Sonix IV model-SS101H 230 (USA) sonicator was used.

## 2.2. Material and solvents

Chemicals were of analytical grade and HPLC grade solvents were used. MKT sodium was provided by Hikma Pharma, Giza district, Egypt (batch number:MT17020021), stored in opaque glass vials. RUP

**Figure 1.** Structural formula for (*a*) RUP fumarate and (*b*) MKT sodium.

fumarate was provided by Mash premiere, Fifth settlement, New Cairo, Egypt. Singulair® tablets contain 10.00 mg MKT (batch no. G23009), produced by Global Nabi Pharmaceuticals, 6th of October, Giza, Egypt, bought from a local pharmacy in Egypt. Hisatrup® tablets each contain 10.00 mg RUP (batch no. M 1044018) produced by Mash premiere, fifth settlement, New Cairo, Egypt, bought from a local pharmacy in Egypt. Laboratory made tablets of RUP and MKT in their commercial ratio 1 : 1 w/w were prepared by mixing these components per one tablet: 10.00 mg of RUP, 10.00 mg of MKT with 15.00 mg lactose, 20.00 mg talc powder, 15.00 mg of maize starch and 10.00 mg magnesium stearate. Methanol, acetonitrile, ethanol and n-propanol were of Sigma-Aldrich products (Germany) HPLC grade, while acetone was from EL-Nasr pharmaceutical chemical co. (ADWIC, Cairo, Egypt) produced at analytical grade. Surfactants like 94% sodium dodecyl sulfate (SDS), carboxy methyl cellulose (CMC), tween 80, cetrimide and β-cyclodextrin (β-CD) were purchased from EL-Nasr pharmaceutical chemical co. (ADWIC, Cairo, Egypt). They were prepared as 0.1% aqueous solutions. Other reagents were also used, such as anhydrous citric acid, disodium hydrogen phosphate, boric acid, sodium hydroxide, sulfuric acid, acetic acid and sodium acetate, which were also purchased from EL-Nasr pharmaceutical chemical co. (ADWIC, Cairo, Egypt). Double-distilled water was also used throughout the whole procedure for both methods. Mcllvaine's buffer covered pH ranging from 2.2 to 6.5 was used by mixing suitable volumes from 0.1 M anhydrous citric acid and 0.20 M disodium hydrogen phosphate. Borate buffer covered pH ranging from 8.5 to 10 was prepared by adding suitable volumes of 0.2 M boric acid and 0.2 M sodium hydroxide; 0.2 M acetate buffer covered pH ranging from 3.50 to 5.50. The buffer was made by mixing appropriate volumes of acetic acid (96%) and sodium acetate trihydrate.

## 2.3. Preparation of standard solution

Stock standard solutions of RUP and MKT were prepared by dissolving 10.0 mg in a 100 ml volumetric flask using methanol. Appropriate dilutions were then carried out with methanol to obtain solutions containing 20.0 µg ml⁻¹. RUP was stable for 3 days without alteration, while MKT must be freshly

**Table 1.** Analytical performance data for the determination of the studied drugs by the proposed methods.

| parameter / drug | method (I) RUP | method (II) RUP | MKT |
|---|---|---|---|
| wavelength (nm) | 277 nm/455 nm | 261 nm | 371 nm |
| linearity range (µg ml$^{-1}$) | 0.20–2.00 | 0.10–4.00 | 0.20–1.60 |
| intercept ($a$) | 29.14 | −0.18 | −0.78 |
| slope ($b$) | 219.54 | 7.44 | 19.38 |
| correlation coefficient ($r$) | 0.9999 | 0.9999 | 0.9999 |
| S.D. of residuals ($S_{y/x}$) | 1.38 | 0.036 | 0.05 |
| S.D. of intercept ($S_a$) | 0.79 | 0.01 | 0.03 |
| S.D. of slope ($S_b$) | 0.88 | 0.01 | 0.04 |
| percentage relative standard deviation, % RSD | 0.574 | 0.99 | 0.82 |
| percentage relative error, % Error | 0.235 | 0.40 | 0.31 |
| limits of detection, LOD (ng ml$^{-1}$) | 59.00 | 31.00 | 66.00 |
| limits of quantitation, LOQ (ng ml$^{-1}$) | 179.00 | 94.00 | 200.00 |

prepared as it was stable for 2 h. Both drugs were protected from light by covering them with aluminium foil, particularly MKT as it is photosensitive.

## 2.4. Procedures

### 2.4.1. Methods for calibration graphs

#### 2.4.1.1. Method I
Aliquots of RUP standard solutions were transferred into a series of 10 ml volumetric flasks, 1 ml of 0.1 M sulfuric acid and 0.80 ml of 0.10% w/v SDS were added and completed to volume with double-distilled water so that the final concentration of RUP is in the linear range (0.20–2.00 µg ml$^{-1}$). The fluorescence intensity was recorded at 277/455 nm. A blank experiment was performed, and the relative fluorescence intensity (RFI) was graphed against the corresponding drug concentrations in µg ml$^{-1}$. The regression equation was then derived.

#### 2.4.1.2. Method II
In a set of 10 ml volumetric flasks, aliquots of RUP or MKT standard solutions covering the studied linear range for each drug (0.10–4.00 µg ml$^{-1}$ for RUP and 0.20–1.60 µg ml$^{-1}$ for MKT) were transferred. One millilitre of McIlvaine's buffer pH 2.60 was added and diluted to the mark with double-distilled water. Synchronous spectra of the solutions were recorded at $\Delta\lambda = 140$ nm and then the first derivative synchronous fluorescence spectra of RUP and MKT were derived using Cary Eclipse software. The peak amplitudes of the first derivative spectra ($^1$D) were recorded at 261 nm for RUP and at 371 nm for MKT. The peak amplitude of the first derivative spectra ($^1$D) was then plotted against the drug concentration in µg ml$^{-1}$ to construct the calibration graph and the corresponding regression equations.

### 2.4.2. Analysis of rupatadine/montelukast synthetic mixtures

Synthetic mixtures of RUP and MKT in the concentration range mentioned in table 1 were prepared from their standard stock solutions in the pharmaceutical ratio 1 : 1. The mixtures were treated as under §2.4.1. The peak amplitudes of the first derivative synchronous spectra ($\Delta^1$D) and the relative accuracy was calculated concurrently for each drug in the same ratios.

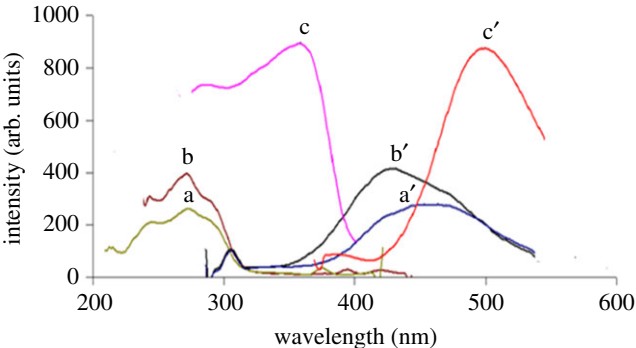

**Figure 2.** Excitation and the emission spectra of a,a′ 1.6 µg ml$^{-1}$ RUP in aqueous solution containing McIlvaine's buffer pH 2.6. b,b′ 1.6 µg ml$^{-1}$ RUP in acidic aqueous solution containing 0.1% w/v SDS. c,c′ 1.6 µg ml$^{-1}$ MKT in aqueous solution containing McIlvaine's buffer pH 2.6.

### 2.4.3. Analysis of pharmaceutical preparations

#### 2.4.3.1. Single tablets

The contents of either ten Hisatrup® tablets or Singulair® tablets were triturated well individually. A 10.0 mg equivalent amount of powder was weighed then added to 100 ml volumetric flasks completed with methanol. Sonication was applied for 30 min and then samples were filtered. Tablet extracts were diluted as appropriate to reach the working range. Then, the general procedure, designated for calibration graphs, was followed. The contents of tablets were computed from the regression equations.

#### 2.4.3.2. Co-formulated tablets

Ten tablets containing RUP and MKT in pharmaceutical ratio 1 : 1 was mixed and a quantity of the powdered tablets equivalent to 10 mg of each drug was moved to a 100 ml volumetric flask. About 80 ml methanol was added and sonicated for 30 min, completed to the mark with the same solvent and filtered. The procedure explained previously was followed. The regression equations corresponding to each of the two drugs were used to determine the content of tablets.

### 2.4.4. Comparison method

The comparison method used to evaluate the obtained results is based on RP-HPLC. The chromatographic separation was achieved on HibarR 250–4, C-18 columns (250 × 4.6 mm, 5 um) using a mobile phase consisting of methanol : water (90 : 10 v/v) with 0.1% triethylamine (pH 3.41 adjusted with ortho phosphoric acid) at a flow rate of 1 ml min$^{-1}$ and a detection wavelength of 260 nm [27].

## 3. Results and discussion

RUP and MKT exhibit native fluorescence at 277/455 nm and 350/450 nm respectively, as presented in figure 2. The emission spectra of different concentrations of RUP in an aqueous acidic solution containing 0.1% w/v SDS at 455.00 nm are illustrated in figure 3.

A great overlap between the emission spectra of RUP and MKT is impossible to be separated by conventional spectrofluorometry (figure 2). So, the SFS technique is a good alternative for improving the selectivity. Different values of Δλ were examined to improve the resolution of this mixture. electronic supplementary material, figure S1 shows that RUP and MKT synchronous fluorescence spectra were overlapped. Hence, their simultaneous quantification and separation is challenging. So, the FDSFS was adopted to estimate the two drugs simultaneously. The fluorescence spectra of RUP and MKT were well resolved with sharp zero-crossing point for each drug (figure 4). RUP could be well calibrated using FDSFS at 261 nm in the presence of MKT (electronic supplementary material, figure S2-A). As shown in the figure, RUP could be quantitated at 261 nm, 266 nm and 275 nm, and the selected wavelength was 261 nm as the readings obtained at zero-crossing points at 266 nm and 275 nm are not reproducible. MKT could be well quantitated at 371 nm in the presence of RUP as shown in electronic supplementary material, figure S2-B.

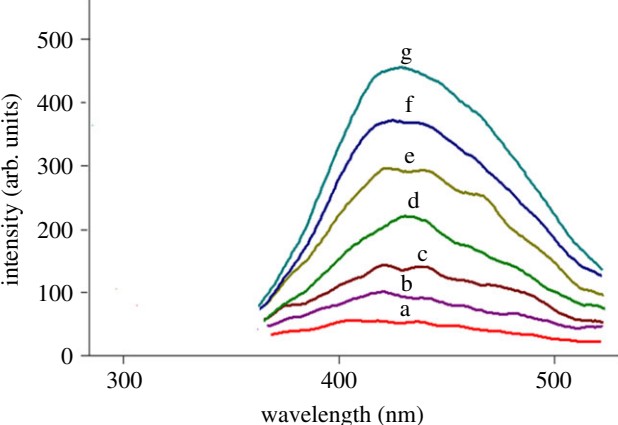

**Figure 3.** Emission spectra of different concentrations of RUP (a–g) (blank, 0.2, 0.4, 0.8, 1.2, 1.6, 2 µg ml$^{-1}$) in acidic aqueous solution containing 0.1% w/v SDS at 455 nm after excitation at 277 nm.

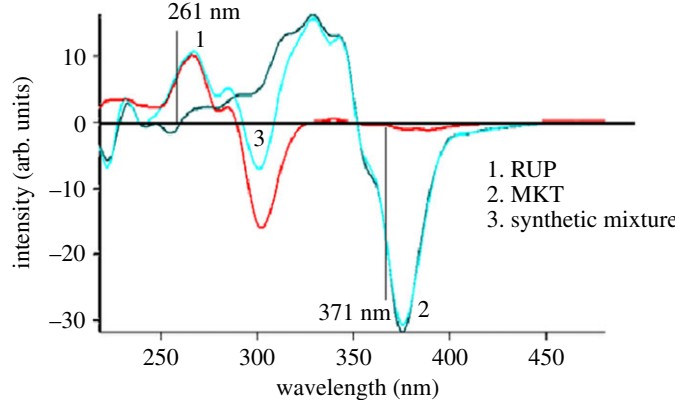

**Figure 4.** First derivative spectroscopy at $\Delta\lambda = 140$ nm for (1) 1.00 µg ml$^{-1}$ RUP. (2) 1.00 µg ml$^{-1}$ MKT. (3) Synthetic mixture of 1 µg ml$^{-1}$ RUP and 1 µg ml$^{-1}$ MKT.

## 3.1. Optimizing the experimental conditions

Factors that may affect the fluorescence intensities for both drugs were studied, by changing one parameter while fixing the others. The results of this study are illustrated in figures 5 and 6

Diluting solvents were examined for better sensitivity including double-distilled water, acetonitrile, ethanol, methanol, n-propanol and acetone but double-distilled water was the best diluting solvent in both methods (figures 5a and 6a). It was observed that RUP suffers from low sensitivity compared to MKT which in turn is characterized by high sensitivity. So, water was selected as the optimum diluting solvent to enhance the fluorescence of RUP. Different buffer systems were studied for both methods including 0.2 M acetate buffer covering a range from 3.5 to 5.5, borate buffer covering a range from 8.5 to 10.5, McIlvaine's buffer covering a range from 2.2 to 6.5, 0.1 N sodium hydroxide for higher pH 12.0 and 0.1 M sulfuric acid pH 1.50 and their volumes were also studied.

For method I, it was clear that the native fluorescence of RUP was enhanced by decreasing pH, so 0.1 M sulfuric acid was chosen as it gave the highest intensity figure 5b. A volume of 1.00 ml of 0.10 M sulfuric acid was selected for this method as it greatly enhanced the intensity of RUP, figure 5c.

For method II, pH was found to have a great effect on the SFS intensity. The FI for MKT is constant from 2.20 up to 5.00 while for RUP, the intensity increases by decreasing pH. Unfortunately, 0.1 M sulfuric acid caused a marked decrease in the fluorescence intensity of MKT. McIlvaine's buffer pH 2.6 was selected as it gave adequate sensitivity for both drugs, figure 6b. Borate buffer solutions have decreased the sensitivity for both drugs. The highest sensitivity for both drugs was achieved using 1 ml of McIlvaine's buffer pH 2.6, figure 6c.

Different surfactants were investigated, including cetrimide, SDS, tween 80, CMC and $\beta$ cyclodextrin.

For method I, SDS was selected as it greatly enhanced the fluorescence intensity of RUP, figure 5d. SDS concentrations such as 0.1–0.5–1.0% w/v were studied and 0.1% w/v SDS was selected for the reproducibility of the results. The volume of SDS was also studied from 0.2–2 ml and 0.80 ml was found to be the best as it gave the highest FI, figure 6e.

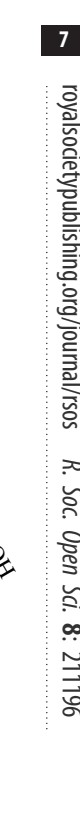

**Figure 5.** (a) The effect of diluting solvents on the native fluorescence of RUP (2 µg ml$^{-1}$). (b) The effect of pH on the native fluorescence of RUP (2 µg ml$^{-1}$). (c) The effect of different volumes of sulfuric acid on the native fluorescence of RUP (2 µg ml$^{-1}$). (d) The effect of different surfactants on the native fluorescence of RUP (2 µg ml$^{-1}$). (e) The effect of different volumes of 0.1% SDS on the native fluorescence of RUP (2 µg ml$^{-1}$).

For method II, no surfactant was chosen for this study as SDS, β-CD and cetrimide significantly decreased the SFI of MKT. Although high FI was obtained for MKT with tween-80, it's not selected as it slightly improved the FI of RUP but markedly enhanced the SFI of MKT and was always over the range (greater than 1000), so it failed in analysing the studied mixture in the pharmaceutical ratio (1 : 1).

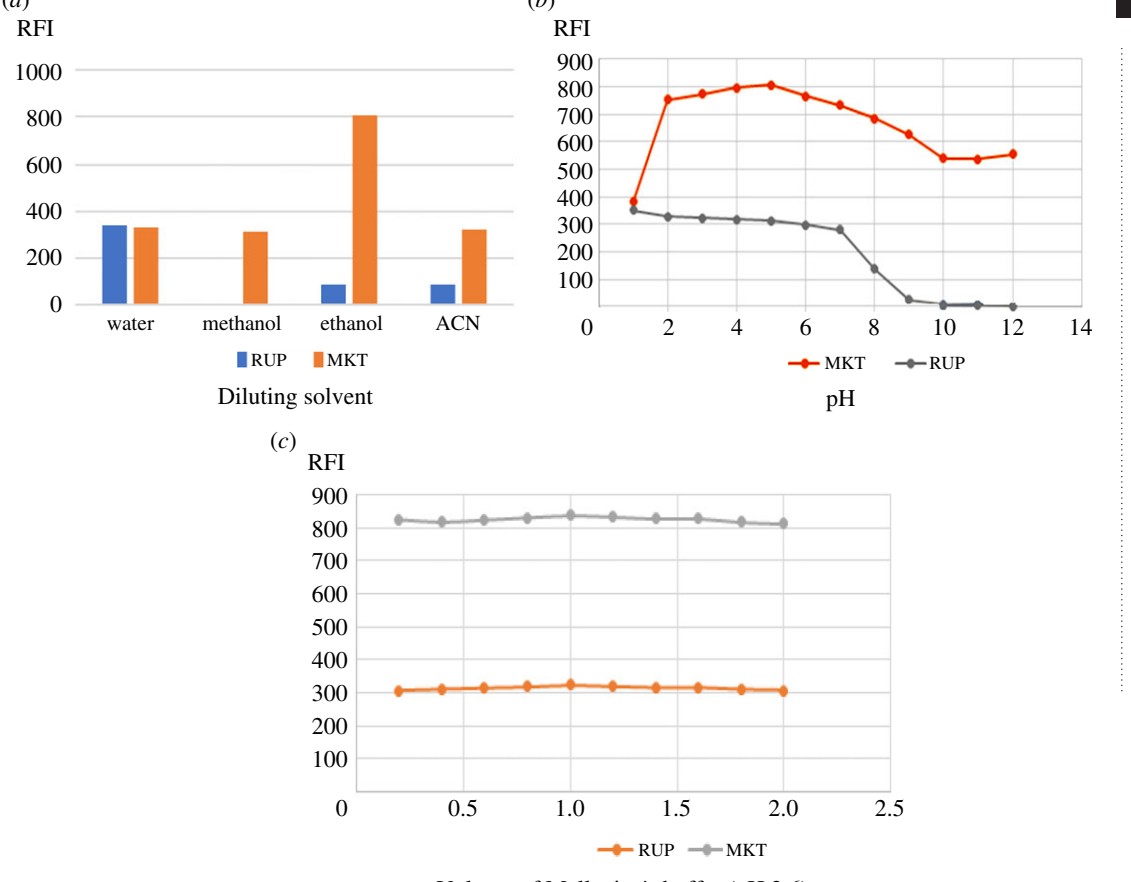

**Figure 6.** (*a*) The effect of diluting solvents on the native fluorescence of both RUP (2 µg ml$^{-1}$) and MKT (1.6 µg ml$^{-1}$). (*b*) The effect of pH showing that pH 2.6 on the native fluorescence of both RUP (2 µg ml$^{-1}$) and MKT (1.6 µg ml$^{-1}$). (*c*) The effect of different volumes of McIlvaine's buffer pH 2.6 on the native fluorescence of both RUP (2 µg ml$^{-1}$) and MKT (1.6 µg ml$^{-1}$).

Varying the value of $\Delta\lambda$ was performed. A direct relationship was found between choosing an optimum value of $\Delta\lambda$ and the sensitivity and resolution in the synchronous fluorescence. To reach the optimal spectra shape, a wide scale of $\Delta\lambda$ (20–200 nm) was examined. The best $\Delta\lambda$ for RUP and MKT was 140 nm, giving well-defined spectra with minimal spectral interference. Smaller or larger values of $\Delta\lambda$ than the ideal one exhibited low SFI and poor separation.

MKT is photosensitive so its stock solution and calibration standards were protected from light by packaging in aluminium foiled flasks and must be freshly prepared as it is stable for 2 h in the refrigerator. RUP is found to be stable for 3 days in the refrigerator.

## 3.2. Validation of the developed methods

Both methods were tested to ensure the validation parameters such as linearity, range, selectivity, specificity, accuracy, precision, LOD and LOQ in accordance with ICH Q2(R1) recommendations [28].

Linear ranges were calculated from the calibration graphs depending on the RFI or $^1$D values with the drug concentrations. The ranges were 0.2–2 µg ml$^{-1}$ for RUP in conventional fluorometric technique (method I), 0.1–4 µg ml$^{-1}$ for RUP at 261 nm and 0.1–1.6 µg ml$^{-1}$ for MKT at 377 nm in the FDSFS technique (method II). Table 1 indicates the results of regression analysis.

The analysis of the data resulted in the regression equations:

$$\text{method I:}\quad \text{RFI} = 29.14 + 219.54 \times (r = 0.9999) \text{ for RUP at 455 nm}$$

and

$$\text{method II}\quad {}^1\text{D} = -0.180 + 7.44 \times (r = 0.9999) \text{ for RUP at 261 nm}$$
$$ {}^1\text{D} = -0.78 + 19.38 \times (r = 0.9999) \text{ for MKT at 371 nm}$$

where RFI = relative fluorescence intensity, $^1$D = peak amplitude of first derivative spectra, X = drug concentrations in µg ml$^{-1}$ for RUP and MKT, $r$ = correlation coefficient.

As per ICH Q2(R1) [27], LOQ could be determined as the lowest concentration that would be measured within the accuracy and precision, while the LOD is the lowest concentration that could be detected. Their values were calculated mathematically as per ICH equations [27] and abridged in table 1.

Accuracy was checked by calculating the per cent recoveries as shown in table 2, by determining the two drugs in the pure and pharmaceutical dosage forms through the referred concentrations. By comparing the results of the studied methods with the comparison methods [27], accuracy was successfully guaranteed.

Intraday and intermediate precision and interday precision were calculated to achieve the repeatability of the proposed methods through calculating their standard deviation (s.d.), mean, relative standard deviation (RSD) and the percentage relative error (% Error) which is calculated by dividing the RSD over the square root of number of samples and then multiplying by 100. The intraday precision was done by getting three different concentrations and measuring them three successive times in the same day, while the interday precision was assessed by measuring these three concentrations in 3 subsequent days. Table 3 shows the values of assessing the precision.

Changing some experimental conditions in both methods was carried out to test robustness. Variations in pH, volume of buffer and surfactant were performed.

For method I, changing the volume of 0.1 M of sulfuric acid and SDS by ±0.20 ml did not affect the FI of RUP. For method II, changing of the volume of Mcllvaine's buffer by ±0.20 ml or the pH 2.6 by ±0.20 did not affect the measured $^1$D of both drugs, illustrated in table 4.

The selectivity was examined by testing the presence of interference from the excipients in the tablets in both methods. No interference was found from lactose, talc or magnesium stearate. In addition, the two drugs could be quantified at the zero crossing of each other without interference.

## 3.3. Applications

### 3.3.1. Analysis of rupatadine/montelukast synthetic mixtures

The proposed synchronous method was used to analyse the two drugs in their 1 : 1 synthetic mixture. Electronic supplementary material, table S1 showed acceptable percentage recoveries for both drugs.

### 3.3.2. Analysis of rupatadine/montelukast combined tablets

The studied drugs in single form or in combined dosage forms were analysed by the two methods. The results were compared with other comparison methods [27] and no significant difference was observed as revealed from student's $t$-test and variance ratio $F$-test [29]. Tables 5 and 6 indicated the data of analysis of different formulations.

### 3.3.3. Assessment of the Green property

Greening an analytical procedure is very challenging in analysis due to the large use of organic solvents. Decreasing the use of these solvents or replacing them with green ones is a method for greenness for any analytical method. Three ways were used to evaluate the greenness of these methods. The green analytical procedure index (GAPI) is a recent method for measuring the greenness, first introduced by Plotka–Wasylka [30]. It follows all the stages of the method starting from the sample collection to the waste treatment. It offers a thorough assessment of each step in the analytical method by including 15 items to be examined using three levels of colour: green, yellow or red. The green profiles for the proposed spectrofluorometric methods using the GAPI tool are presented in table 7 and electronic supplementary material, table S2. MKT must be kept in aluminium foil and in the refrigerator under normal conditions, so the fourth parameter was yellow shaded in method II. The 5th parameter is shaded yellow as there was a bit sample preparation as filtration in both methods. The two pictograms (10,11) related to the reagents and solvents were yellow shaded; due to the use of some hazardous chemicals, such as methanol, sulfuric acid, SDS and Mcllvaine's buffer pH 2.6 even their usage by small volume therefore GAPI evaluation may oppress some methods. The amount of waste was between 1 to 10 ml shaded yellow, with no treatment of the waste indicated by red shading, covering field no. 15 in both methods.

**Table 2.** Application of the proposed methods for the determination of the studied drugs in raw materials. The tabulated t- and F-values at 2.77 and 19 at p = 0.05, respectively [29].

| parameters | method I | | | method II | | | | | | comparison method [27] | | | |
|---|---|---|---|---|---|---|---|---|---|---|---|---|---|
| | RUP at 455 nm | | | RUP at 261 nm | | | MKT at 371 nm | | | RUP | | MKT | |
| | amount taken µg ml⁻¹ | amount found µg ml⁻¹ | % found[a] | amount taken µg ml⁻¹ | amount found µg ml⁻¹ | % found[a] | amount taken µg ml⁻¹ | amount found µg ml⁻¹ | % found[a] | amount taken µg ml⁻¹ | % found[a] | amount taken µg ml⁻¹ | % found[a] |
| | 0.20 | 0.200 | 100.00 | 0.10 | 0.099 | 99.00 | 0.20 | 0.197 | 98.50 | 20.00 | 100.68 | 20.00 | 101.11 |
| | 0.40 | 0.404 | 100.75 | 0.20 | 0.196 | 98.00 | 0.40 | 0.402 | 100.50 | 30.00 | 99.18 | 30.00 | 98.51 |
| | 0.80 | 0.797 | 99.50 | 0.40 | 0.398 | 99.50 | 0.80 | 0.804 | 100.50 | 40.00 | 100.29 | 40.00 | 100.57 |
| | 1.20 | 1.202 | 100.00 | 1.00 | 1.009 | 100.90 | 1.20 | 1.203 | 100.25 | | | | |
| | 1.60 | 1.590 | 99.75 | 2.00 | 3.998 | 99.90 | 1.30 | 1.299 | 99.92 | | | | |
| | 2.00 | 2.008 | 100.20 | 4.00 | 3.999 | 99.98 | 1.40 | 1.386 | 99.00 | | | | |
| | | | | | | | 1.60 | 1.609 | 99.92 | | | | |
| mean | | | 100.09 | | | 99.55 | | | 99.89 | | 100.05 | | 100.06 |
| ±S.D. | | | 0.58 | | | 0.98 | | | 0.82 | | 0.78 | | 1.32 |
| % RSD | | | 0.57 | | | 0.99 | | | 0.82 | | 0.78 | | 1.32 |
| % error | | | 0.24 | | | 0.40 | | | 0.31 | | 0.45 | | 0.79 |
| N | | | 6.00 | | | 6.00 | | | 7.00 | | | | |
| N | | | 3.00 | | | | | | | | | | |
| t-test | | | 0.05 | | | 0.08 | | | 0.19 | | | | |
| F-value | | | 1.53 | | | 1.16 | | | 1.41 | | | | |

[a]mean of three determinations.

**Table 3.** Precision data for the estimation of studied drugs by the proposed methods. N.B. Each result is the average of three separate determinations.

| parameters drugs (µg ml$^{-1}$) | | method (I) | | | method (II) | | | | | |
| --- | --- | --- | --- | --- | --- | --- | --- | --- | --- | --- |
| | | RUP | | | RUP | | | MKT | | |
| | | 0.20 | 0.80 | 2.00 | 0.40 | 1.00 | 4.00 | 0.20 | 1.00 | 1.60 |
| intraday | mean | 100.10 | 100.00 | 100.00 | 99.8 | 99.99 | 100.00 | 99.60 | 100.00 | 100.00 |
| | ± S.D. | 1.57 | 0.87 | 0.61 | 1.49 | 1.84 | 1.90 | 1.65 | 0.77 | 0.80 |
| | % RSD | 1.57 | 0.866 | 0.61 | 1.50 | 1.84 | 1.90 | 1.66 | 0.78 | 0.80 |
| | % error | 0.91 | 0.50 | 0.35 | 0.86 | 1.06 | 1.01 | 0.96 | 0.45 | 0.46 |
| interday | mean | 99.33 | 100.00 | 99.99 | 99.63 | 100.00 | 100.00 | 100.20 | 99.56 | 101.03 |
| | ± S.D. | 1.03 | 0.53 | 0.80 | 0.81 | 0.90 | 0.40 | 0.87 | 1.37 | 0.61 |
| | % RSD | 1.04 | 0.53 | 0.80 | 0.811 | 0.90 | 0.40 | 0.87 | 1.38 | 0.61 |
| | % error | 0.60 | 0.31 | 0.46 | 0.47 | 0.52 | 0.23 | 0.50 | 0.80 | 0.35 |

**Table 4.** Robustness testing of the developed methods.

| parameters | mean ± s.d. | % RSD |
|---|---|---|
| method I | | |
| 1 - volume of 0.1 M sulfuric acid 1 ml ± 0.2 | 100.003 ± 0.69 | 0.69 |
| 2 - volume of 0.1% SDS 0.8 ml ± 0.2 | 100.00 ± 0.17 | 0.17 |
| method II | | |
| 1 - changing in McIlvaine's buffer pH 2.6 ± 0.2 | 100.003 ± 0.66 | 0.66 |
| 2 - volume of McIlvaine's buffer pH 2.6 (1 ± 0.2 ml) | 99.99 ± 0.35 | 0.35 |

**Table 5.** Application of the proposed methods to determine RUP and MKT in prepared combined tablets. The tabulated $t$- and $F$- values at 2.77 and 19 at $p = 0.05$, respectively [29].

| parameter | proposed method | | | comparison method [27] | | |
|---|---|---|---|---|---|---|
| | amount taken ($\mu$g ml$^{-1}$) | amount found ($\mu$g ml$^{-1}$) | percentage found[a] | amount taken ($\mu$g ml$^{-1}$) | amount found ($\mu$g ml$^{-1}$) | percentage found[a] |
| RUP | 0.80 | 0.812 | 101.50 | 20.00 | 20.135 | 100.68 |
| | 1.00 | 0.984 | 98.40 | 30.00 | 29.754 | 99.18 |
| | 1.40 | 1.405 | 100.36 | 40.00 | 40.117 | 100.29 |
| mean | | | 100.09 | | | 100.05 |
| ± S.D. | | | 1.57 | | | 0.78 |
| % RSD | | | 1.57 | | | 0.78 |
| % error | | | 0.91 | | | 0.45 |
| t-test | | | 0.03 | | | |
| F-value | | | 4.05 | | | |
| MKT | 0.80 | 0.802 | 101.00 | 20.00 | 20.221 | 101.11 |
| | 1.00 | 0.997 | 99.30 | 30.00 | 29.552 | 98.51 |
| | 1.40 | 1.401 | 100.14 | 40.00 | 40.226 | 100.57 |
| mean | | | 100.14 | | | 100.06 |
| ± S.D. | | | 0.85 | | | 1.37 |
| % RSD | | | 0.85 | | | 1.37 |
| % error | | | 0.49 | | | 0.79 |
| t-test | | | 0.08 | | | |
| F-value | | | 2.60 | | | |

[a]N.B. mean of three determinations.

Analytical eco scale is another quantitative tool for assessment that is published by Van-Aken *et al.* [31]. Ranking the greenness of the method depends on the penalty point score. The score of penalty points is recorded according to the signal word and no. of pictograms presented in 'The Globally Harmonized System of Classification and Labelling of Chemicals' (GHS) and the safety label data sheet for each chemical or solvent and then subtracted from 100. Excellent green methods scored 75 or more but acceptable green methods scored 50 or more as shown in table 7. The conventional method scored 88 while the synchronous method scored 89. Both methods are excellent regarding the analytical eco scale criteria. The penalty points were calculated from the national fire protection association [32].

The last qualitative tool is the old one called the national environmental method index (NEMI) [33]. It describes the greenness through a pictogram divided into four quadrants as in table 7 in which the first

**Table 6.** Determination of RUP and MKT in pharmaceutical preparations using the proposed methods. The tabulated $t$ and $F$-values at 2.77 and 19 at $p = 0.05$, respectively [29].

| parameter | proposed method | | | comparison method [27] | | |
|---|---|---|---|---|---|---|
| | amount taken ($\mu$g ml$^{-1}$) | amount found ($\mu$g ml$^{-1}$) | percentage found[a] | amount taken ($\mu$g ml$^{-1}$) | amount found ($\mu$g ml$^{-1}$) | percentage found[a] |
| **method (I)** | | | | | | |
| Hisatrup® tablets | 0.80 | 0.804 | 100.50 | 20.00 | 20.135 | 100.68 |
| RUP (10.00 mg) | 1.00 | 0.996 | 99.60 | 30.00 | 29.754 | 99.18 |
| | 1.40 | 1.405 | 100.14 | 40.00 | 40.117 | 100.29 |
| mean | | | 100.01 | | | 100.05 |
| ± S.D. | | | 0.56 | | | 0.78 |
| % RSD | | | 0.56 | | | 0.78 |
| % error | | | 0.32 | | | 0.45 |
| t- test | | | 0.05 | | | |
| F-value | | | 2.95 | | | |
| **method (II)** | | | | | | |
| Hisatrup® tablets | 0.80 | 0.804 | 100.5 | 20.00 | 20.135 | 100.68 |
| RUP (10.00 mg) | 1.00 | 0.994 | 99.40 | 30.00 | 29.754 | 99.18 |
| | 1.40 | 1.402 | 100.14 | 40.00 | 40.117 | 100.29 |
| mean | | | 100 | | | 100.05 |
| ± S.D. | | | 0.56 | | | 0.78 |
| % RSD | | | 0.56 | | | 0.78 |
| % error | | | 0.32 | | | 0.45 |
| t-test | | | 0.06 | | | |
| F-value | | | 1.92 | | | |
| **method (II)** | | | | | | |
| Singulair® tablets | 0.80 | 0.804 | 100.50 | 20.00 | 20.221 | 101.11 |
| MKT (10.00 mg) | 1.00 | 0.994 | 99.40 | 30.00 | 29.552 | 98.51 |
| | 1.40 | 1.402 | 100.14 | 40.00 | 40.226 | 100.57 |
| mean | | | 100.01 | | | 100.06 |
| ± S.D. | | | 0.56 | | | 1.37 |
| % RSD | | | 0.56 | | | 1.37 |
| % error | | | 0.32 | | | 0.79 |
| t-test | | | 0.05 | | | |
| F-value | | | 5.98 | | | |

[a]N.B. mean of three determinations.

quadrant shows reagents that are not persistent, bio-accumulative or toxic. The second one (Hazardous) includes reagents that are not hazard; the third one, called corrosive, includes pH less than 2 and more than 12 while the last quadrant, called waste, includes overall waste less than 50 gm or 50 ml. The conventional developed method I fulfills NEMI criteria as the first and fourth quadrants were green while in the second and third quadrants the used pH is 1.5 which deviated from the selected range due to the usage of the sulfuric acid which was considered corrosive and hazardardous, while the synchronous method successfully fulfills NEMI criteria as all quadrants are green. One quick look will indicate whether the method is green or not.

It is obvious that the developed methods are well matched with the three green analytical chemistry tools which shows that these methods are eco-friendly; moreover, they are simple, sensitive and rapid.

**Table 7.** Results for the evaluation of the developed conventional method by the three green chemistry tools (method I &II).

1 - green analytical procedure index (GAPI)

| method I | method II |
|---|---|
| | |

2 - analytical Eco scale score

A - method I

reagents/instruments

| reagent, volume (ml) | no. pictograms | word sign [31] | penalty points |
|---|---|---|---|
| methanol, 1 ml | 3 | danger | 6 |
| 0.1% SDS, 0.8 ml | 1 | warning | 1 |
| 0.1 M sulfuric acid, 1 ml | 1 | danger | 2 |
| water | | | 0 |

| item | | | penalty points |
|---|---|---|---|
| spectrofluorometer | <0.1 k w h per sample | | 0 |
| waste | no treatment | | 3 |
| occupational hazards | analytical process hermitization | | 0 |
| total penalty points | | | $\sum$ 12 |
| analytical eco scale score | | | $100 - 12 = 88$ |

B - method II

reagents/instruments

| reagent, volume (ml) | no. Pictograms | word Sign [31] | penalty points |
|---|---|---|---|
| methanol, 1 ml | 3 | danger | 6 |
| McIlvaine's buffer pH 2.6, 1 ml | 2 | warning | 2 |
| water | | | 0 |

| **item** | | | penalty points |
|---|---|---|---|
| spectrofluorometer | <0.1 k w h per sample | | 0 |
| waste | no treatment | | 3 |
| occupational hazards | analytical process hermitization | | 0 |
| total penalty points | | | $\sum$ 11 |
| analytical eco scale score | | | $100 - 11 = 89$ |

3 - NEMI pictogram

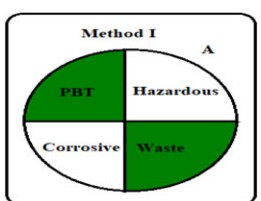

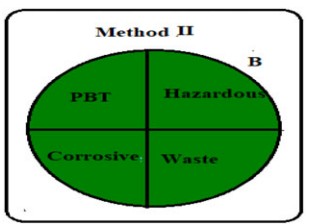

# 4. Conclusion

A green, simple and highly sensitive conventional fluorometric method is established to quantify RUP in pharmaceutical dosage forms. Moreover, a FDSFS is used as a simple, selective and green technique to determine RUP and MKT in pure form and in their pharmaceuticals. The two methods are validated according to pharmacopeial guidelines. Owing to the simplicity and sensitivity of the proposed methods, they can be an excellent alternative to other sophisticated techniques in quality control units.

Data accessibility. Data are available from the Dryad Digital Repository: https://doi.org/10.5061/dryad.kwh70rz43 [34].
Authors' contributions. R.G. carried out the laboratory work, participated in data analysis and participated in the design of the study; M.I.E. and M.M.T. drafted the manuscript, carried out the statistical analyses, conceived of the study and followed up the experimental work; F.I. coordinated the study participated in data analysis and helped draft the manuscript. All authors gave final approval for publication.
Competing interests. We declare we have no competing interests.
Funding. We received no funding for this study.

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
