## [Peer Review File · Royal Society Open Science]

Review History

RSOS-211196.R0 (Original submission)

Review form: Reviewer 1

Is the manuscript scientifically sound in its present form?

Yes

Are the interpretations and conclusions justified by the results?

Yes

Is the language acceptable?

Yes

Do you have any ethical concerns with this paper?

No

Have you any concerns about statistical analyses in this paper?

Yes

Recommendation?

Accept with minor revision (please list in comments)

Comments to the Author(s)

This is a fine contribution and can be accepted for publication. Only language should be carefully revised for any mistakes.

Review form: Reviewer 2**Is the manuscript scientifically sound in its present form?**

Yes

Are the interpretations and conclusions justified by the results?

Yes

Is the language acceptable?

Yes

Do you have any ethical concerns with this paper?

No

Have you any concerns about statistical analyses in this paper?

No

Recommendation?

Accept with minor revision (please list in comments)

Comments to the Author(s)

In this manuscript, the authors developed a green developed green quantitative methods for the quantitative spectrofluorimetric analysis of rupertadine and montelukast at nanogram scale using direct and synchronous techniques. The reaction product showed an acceptable sensitivity and selectivity towards the targeted analytes in a mixture. I agree with the scientific quality of this research. I think the manuscript is suitable for publication in the Royal Society Open Science journal pending revision.

- Through the methods "sensitivity indicator", the calculated LOQ is 36 ng/mL for RUP, respectively 13.4, and 15 ng/mL for RUP and MKT, Why the constructed calibration curves away from these LOQ values so much as they began from 100 and/or 200 ng/mL???

- The synthetic laboratory-made tablets of RUP and MKT in their commercial ratio 1:1 w/w, were prepared by mixing these components per one tablet: 10.00 mg of RUP, 10.00 mg of MKT with 15.00 mg lactose, 20.00 mg talc powder, etc., kindly put the reference guides these manipulations.

- RUP and MKT have single marketed dosage forms, why is the necessity for the lab. Preparation mixture?? Is not marketed in Egypt??

- In Figure 2, 3, the emission of 1.0 µg/mL of acidic solution and SDS of RUP gave emission intensity around 400 FI, where the concentration of 1.2 µg/mL gives FI of 200, explain??

- In the greenness study part, we noted that:

1. For the 1st procedure Green analytical procedure index (GAPI)

- The author mentioned that there is NO extraction step (item NO 6), but what about adding 100 methanol, then sonicate for 30 min. (which required more energy exhaust)?? As well as in item NO 7, the original standard stock solutions were prepared in 100 mL methanol, I think it should be added to the scale of GAPI.

- Part reagents and solvents: the amount of the used solvent, the stock solution prepared in 100 mL methanol for both drugs, moreover, and due its instability, MNT should be prepared every three days, in the same exhausted solvent. The wastes contain organic solvent as well as calculated each time to be involved in the GAPI scale too.
 - Instrumentation and waste's part: the actual volume of the wastes is calculated as the total repeated sample (as for example 10 mL each sample, containing NO of organic solvent, and reagent, multiplied in the whole repeated samples in the same ruing day), as in case of validation steps.
 - This should be considered as well as in the case of method II, table S2-B.
2. In the calculation of the Analytical Eco scale score
 - Methanol usage should be calculated for more than 100 mL, not only 1 mL.
 3. In the last metric method related to NEMI based on evaluation of TRI and/or RCRA lists:
 - A 100 mL methanol, 1 mL of 0.1 M sulfuric acid (corrosive, and contained in the waste), that will affect, and they present in the TRI list as well as in the RCRA, and unfortunately, they will affect the quadrant of hazards as well, so the greenness profile using NEM index, for both methods should be revised to the real values.
 - Finally, it is better to add further discussion to clarify the obtained results. Other miswriting and suggestions are present throughout the body of the pdf, kindly refer and reply.

Review form: Reviewer 3

Is the manuscript scientifically sound in its present form?

Yes

Are the interpretations and conclusions justified by the results?

Yes

Is the language acceptable?

No

Do you have any ethical concerns with this paper?

No

Have you any concerns about statistical analyses in this paper?

No

Recommendation?

Major revision is needed (please make suggestions in comments)

Comments to the Author(s)

The manuscript describes the development and validation of two spectrofluorimetric methods for determination of rupertadin alone or in combination with montelukast in pharmaceutical formulations. Additionally, assessment of the 'green' character of developed methods was performed. In my opinion, the paper can be published after the authors address the issues in the below comments.

Comments:

1. English language should be substantially improved. There are many syntax and grammatical errors, mixing of tenses, etc. The text is difficult to understand at certain points. The manuscript is

recommended to be read and corrected by a native English speaking scientist. Also, avoid capitalizing the chemical names.

2. Units: standard abbreviation for “hour” is h. There should be a space between number and unit (including M). Please be careful not to allow for a number and its unit in two subsequent rows. Use the nonbreaking space.

3. Title: perhaps the term “nanogram scale” is a slight exaggeration. LOD of methods were at the nanogram level, but not the linear range.

4. Introduction, penultimate paragraph: I found two papers on HPLC methods for simultaneous determination of rupatadine and montelukast in pharmaceutical formulations. Authors cited only one, the other is missing:

Redasani, VK, Kothawade, AR, Surana, SJ, J. Anal. Chem. 2014, 69, 384-389
DOI10.1134/S1061934814040121

5. Experimental, subchapters 2.1 and 2.2: it is very unusual to give materials and equipment as bullet points. Please check the prescribed form of the journal in the Instructions for authors. Have you really used distilled water or perhaps deionized water? Mode of preparation should be given. For all chemicals, instruments and other equipment, the producer should be given. It is now missing in several instances.

6. Experimental, subchapter 2.4.1: the actual concentrations of analytes in standard solutions for calibration should be given for both methods. Line 47/48: what is meant by “plotted against the 1 drug concentration”? What is “1 drug”?

7. For simultaneous determination of both analytes, authors prepared only mixture 1:1. What about other ratios? Would it be possible to use Method II to determine both analytes at extreme ratios, i.e. if one of them was in great excess? Have authors checked that?

8. Chapter 3.1: some results of methods optimization in the form of tables or graphs should be given.

9. Validation of the developed methods & Table 1: LOQ is normally the lowest point of the calibration range, but in both methods they are much lower than the lowest point. Therefore, authors should further explain this discrepancy. How is the “percentage relative error” in Table 1 calculated?

10. Table 2 and the corresponding text on page 10: different references ([6] or [7]) are given in text and in Table 2 for the comparison method used to determine accuracy. In any case, the comparison method should be briefly described in the Experimental part. What is meant by “Error” in Table 2 as there are no results for this entry?

11. Table 3 and the corresponding text on page 10: What is meant by “% Error” in Table 3 and how was it calculated? There’s no explanation in the text.

12. Robustness testing (page 10): where are the results of testing? Some are given in the next paragraph, but most are missing.

13. Selectivity testing (page 10): no results are shown, Moreover, there are many more excipients that could be used in the preparation of pharmaceutical formulations. Also, it would be good to see if the quantification of both analytes is possible at ratios different from 1:1.

14. Tables 4 and 5 and the corresponding text on page 11: again, discrepancy in references for comparison method in the text and in Tables. What is meant by “% Error” and how was it calculated?

15. Table 6: at GAPI pictograms, Table 7 is mentioned which is missing. What is meant by “hermitization” – probably hermetization?

16. Finally, I would suggest that the results of developed methods should be compared to results of a rather different method (HPLC) that is considered more selective than derivative spectrometry used by authors for comparison.

Decision letter (RSOS-211196.R0)

Dear Dr El-Awady:

Title: Green quantitative spectrofluorimetric analysis of rupatadine and montelukast at nanogram scale using direct and synchronous techniques.

Manuscript ID: RSOS-211196

The editor assigned to your manuscript has now received comments from reviewers. We would like you to revise your paper in accordance with the referee and Subject Editor suggestions which can be found below (not including confidential reports to the Editor). Please note this decision does not guarantee eventual acceptance.

Please submit your revised paper before 17-Sep-2021. Please note that the revision deadline will expire at 00.00am on this date. If we do not hear from you within this time then it will be assumed that the paper has been withdrawn. In exceptional circumstances, extensions may be possible if agreed with the Editorial Office in advance. We do not allow multiple rounds of revision so we urge you to make every effort to fully address all of the comments at this stage. If deemed necessary by the Editors, your manuscript will be sent back to one or more of the original reviewers for assessment. If the original reviewers are not available we may invite new reviewers.

Yours sincerely,
Dr Ellis Wilde
Publishing Editor, Journals

RSC Associate Editor
Comments to the Author:
(There are no comments.)

RSC Subject Editor
Comments to the Author:
(There are no comments.)

Reviewers' Comments to Author:

Reviewer: 1

Comments to the Author(s)

This is a fine contribution and can be accepted for publication. Only language should be carefully revised for any mistakes.

Reviewer: 2

Comments to the Author(s)

In this manuscript, the authors developed a green developed green quantitative methods for the quantitative spectrofluorimetric analysis of rupertadine and montelukast at nanogram scale using direct and synchronous techniques. The reaction product showed an acceptable sensitivity and selectivity towards the targeted analytes in a mixture. I agree with the scientific quality of this research. I think the manuscript is suitable for publication in the Royal Society Open Science journal pending revision.

- Through the methods "sensitivity indicator", the calculated LOQ is 36 ng/mL for RUP, respectively 13.4, and 15 ng/mL for RUP and MKT, Why the constructed calibration curves away from these LOQ values so much as they began from 100 and/or 200 ng/mL???

- The synthetic laboratory-made tablets of RUP and MKT in their commercial ratio 1:1 w/w, were prepared by mixing these components per one tablet: 10.00 mg of RUP, 10.00 mg of MKT with 15.00 mg lactose, 20.00 mg talc powder, etc., kindly put the reference guides these manipulations.

- RUP and MKT have single marketed dosage forms, why is the necessity for the lab. Preparation mixture?? Is not marketed in Egypt??

- In Figure 2, 3, the emission of 1.0 $\mu\text{g}/\text{mL}$ of acidic solution and SDS of RUP gave emission intensity around 400 FI, where the concentration of 1.2 $\mu\text{g}/\text{mL}$ gives FI of 200, explain??
- In the greenness study part, we noted that:
 1. For the 1st procedure Green analytical procedure index (GAPI)
 - The author mentioned that there is NO extraction step (item NO 6), but what about adding 100 mL methanol, then sonicate for 30 min. (which required more energy exhaust)?? As well as in item NO 7, the original standard stock solutions were prepared in 100 mL methanol, I think it should be added to the scale of GAPI.
 - Part reagents and solvents: the amount of the used solvent, the stock solution prepared in 100 mL methanol for both drugs, moreover, and due its instability, MNT should be prepared every three days, in the same exhausted solvent. The wastes contain organic solvent as well as calculated each time to be involved in the GAPI scale too.
 - Instrumentation and waste's part: the actual volume of the wastes is calculated as the total repeated sample (as for example 10 mL each sample, containing NO of organic solvent, and reagent, multiplied in the whole repeated samples in the same ruing day), as in case of validation steps.
 - This should be considered as well as in the case of method II, table S2-B.
 2. In the calculation of the Analytical Eco scale score
 - Methanol usage should be calculated for more than 100 mL, not only 1 mL.
 3. In the last metric method related to NEMI based on evaluation of TRI and/or RCRA lists:
 - A 100 mL methanol, 1 mL of 0.1 M sulfuric acid (corrosive, and contained in the waste), that will affect, and they present in the TRI list as well as in the RCRA, and unfortunately, they will affect the quadrant of hazards as well, so the greenness profile using NEM index, for both methods should be revised to the real values.
 - Finally, it is better to add further discussion to clarify the obtained results. Other miswriting and suggestions are present throughout the body of the pdf, kindly refer and reply.

Reviewer: 3

Comments to the Author(s)

The manuscript describes the development and validation of two spectrofluorimetric methods for determination of rupatadin alone or in combination with montelukast in pharmaceutical formulations. Additionally, assessment of the 'green' character of developed methods was performed. In my opinion, the paper can be published after the authors address the issues in the below comments.

Comments:

1. English language should be substantially improved. There are many syntax and grammatical errors, mixing of tenses, etc. The text is difficult to understand at certain points. The manuscript is recommended to be read and corrected by a native English speaking scientist. Also, avoid capitalizing the chemical names.
2. Units: standard abbreviation for "hour" is h. There should be a space between number and unit (including M). Please be careful not to allow for a number and its unit in two subsequent rows. Use the nonbreaking space.
3. Title: perhaps the term "nanogram scale" is a slight exaggeration. LOD of methods were at the nanogram level, but not the linear range.
4. Introduction, penultimate paragraph: I found two papers on HPLC methods for simultaneous determination of rupatadine and montelukast in pharmaceutical formulations. Authors cited only one, the other is missing:

Redasani, VK, Kothawade, AR, Surana, SJ, J. Anal. Chem. 2014, 69, 384-389
DOI10.1134/S1061934814040121

5. Experimental, subchapters 2.1 and 2.2: it is very unusual to give materials and equipment as bullet points. Please check the prescribed form of the journal in the Instructions for authors. Have you really used distilled water or perhaps deionized water? Mode of preparation should be given. For all chemicals, instruments and other equipment, the producer should be given. It is now missing in several instances.

6. Experimental, subchapter 2.4.1: the actual concentrations of analytes in standard solutions for calibration should be given for both methods. Line 47/48: what is meant by "plotted against the drug concentration"? What is "I drug"?

7. For simultaneous determination of both analytes, authors prepared only mixture 1:1. What about other ratios? Would it be possible to use Method II to determine both analytes at extreme ratios, i.e. if one of them was in great excess? Have authors checked that?

8. Chapter 3.1: some results of methods optimization in the form of tables or graphs should be given.

9. Validation of the developed methods & Table 1: LOQ is normally the lowest point of the calibration range, but in both methods they are much lower than the lowest point. Therefore, authors should further explain this discrepancy. How is the "percentage relative error" in Table 1 calculated?

10. Table 2 and the corresponding text on page 10: different references ([6] or [7]) are given in text and in Table 2 for the comparison method used to determine accuracy. In any case, the comparison method should be briefly described in the Experimental part. What is meant by "Error" in Table 2 as there are no results for this entry?

11. Table 3 and the corresponding text on page 10: What is meant by "% Error" in Table 3 and how was it calculated? There's no explanation in the text.

12. Robustness testing (page 10): where are the results of testing? Some are given in the next paragraph, but most are missing.

13. Selectivity testing (page 10): no results are shown, Moreover, there are many more excipients that could be used in the preparation of pharmaceutical formulations. Also, it would be good to see if the quantification of both analytes is possible at ratios different from 1:1.

14. Tables 4 and 5 and the corresponding text on page 11: again, discrepancy in references for comparison method in the text and in Tables. What is meant by "% Error" and how was it calculated?

15. Table 6: at GAPI pictograms, Table 7 is mentioned which is missing. What is meant by "hermitization" - probably hermetization?

16. Finally, I would suggest that the results of developed methods should be compared to results of a rather different method (HPLC) that is considered more selective than derivative spectrometry used by authors for comparison.

Author's Response to Decision Letter for (RSOS-211196.R0)

See Appendix A.

RSOS-211196.R1 (Revision)

Review form: Reviewer 1

Is the manuscript scientifically sound in its present form?

Yes

Are the interpretations and conclusions justified by the results?

Yes

Is the language acceptable?

Yes

Do you have any ethical concerns with this paper?

No

Have you any concerns about statistical analyses in this paper?

No

Recommendation?

Accept as is

Comments to the Author(s)

Authors responded to all comments. manuscript accepted.

Review form: Reviewer 2

Is the manuscript scientifically sound in its present form?

Yes

Are the interpretations and conclusions justified by the results?

Yes

Is the language acceptable?

Yes

Do you have any ethical concerns with this paper?

No

Have you any concerns about statistical analyses in this paper?

No

Recommendation?

Accept as is

Comments to the Author(s)

There NO any comment.

I appreciate their polite replays and as they accepted my sent considerations for the benefit of the manuscript.

Decision letter (RSOS-211196.R1)

Dear Dr El-Awady:

Title: Green quantitative spectrofluorometric analysis of rupatadine and montelukast at nanogram scale using direct and synchronous techniques.

Manuscript ID: RSOS-211196.R1

It is a pleasure to accept your manuscript in its current form for publication in Royal Society Open Science. The chemistry content of Royal Society Open Science is published in collaboration with the Royal Society of Chemistry.

Yours sincerely,
Dr Ellis Wilde
Publishing Editor, Journals

RSC Associate Editor
Comments to the Author:
(There are no comments.)

RSC Subject Editor
Comments to the Author:
(There are no comments.)

Reviewer(s)' Comments to Author:
Reviewer: 1
Comments to the Author(s)
Authors responded to all comments. manuscript accepted.

Reviewer: 2
Comments to the Author(s)
There NO any comment.
I appreciate their polite replays and as they accepted my sent considerations for the benefit of the manuscript.

Appendix A

Response to Reviewers' Comments

Thanks to the reviewers for their thorough reviews and comments which were generally in target and helped to improve the manuscript. All the comments were carefully considered in preparing the revised version and a point-to-point response for these comments can be outlined as follows:

REVIEWER 1

Comment: This is a fine contribution and can be accepted for publication. Only language should be carefully revised for any mistakes.

Reply:

As recommended by the reviewer, the language has been carefully revised throughout the whole manuscript.

REVIEWER 2

Comment 1: Through the methods “sensitivity indicator”, the calculated LOQ is 36 ng/mL for RUP, respectively 13.4, and 15 ng/mL for RUP and MKT, Why the constructed calibration curves away from these LOQ values so much as they began from 100 and/or 200 ng/mL???

Reply:

According to ICH Harmonized Tripartite Guidelines, Validation of Analytical Procedures: Text and Methodology Q2(R1) (<https://www.ich.org/page/quality-guidelines>), LOQ is defined as: “The lowest amount of analyte in a sample which can be quantitatively determined with suitable precision and accuracy” and can be calculated as follows:

$$\text{LOQ} = 10 \sigma / S$$

where S = the slope of the calibration curve and σ can be calculated by a variety of ways, such as calculation based on the standard deviation of the y-intercept of the regression line or calculation based on the standard deviation of blank. In the current study, LOQ was calculated based on the standard deviation of the y-intercept of the regression line.

Following the recommendation of the reviewer, the values of LOQ and LOD have been recalculated using the second approach based on the standard deviation of blank and the results obtained are more acceptable than that of the first method. The new values have been added to Table 1.

Comment 2: The synthetic laboratory-made tablets of RUP and MKT in their commercial ratio 1:1 w/w, were prepared by mixing these components per one tablet: 10.00 mg of RUP, 10.00 mg of MKT with 15.00 mg lactose, 20.00 mg talc powder, etc., kindly put the reference guides these manipulations.

Reply:

The mentioned components were selected as representative examples of common tablet excipients. Lactose and maize starch are both examples of fillers and binders, talc powder is an example of glidants, and magnesium stearate is an example of lubricants. There is no specific reference for a fixed ratio of these tablet excipients as that depends on several factors such as the dose of active ingredients, total tablet weight, tablet disintegration, adsorption effects etc. Moreover, most of the pharmaceutical companies does not mention the nature and amount of tablet excipients in the internal leaflet of their tablet preparations. Therefore, these components were selected just to study any possible interference of common tablet excipients to prove the selectivity of the developed method.

Comment 3: RUP and MKT have single marketed dosage forms, why is the necessity for the lab. Preparation mixture?? Is not marketed in Egypt??

Reply:

The study included the analysis of laboratory-prepared mixtures because RUP and MKT are co-formulated in tablet preparations to give better therapeutic effect especially in treating the symptoms of allergic rhinitis. This combined tablet preparation is not available in the Egyptian market but it is marketed in other countries such as India under the trade names: Rupanex M[®], Montyrup[®].

Comment 4: In Figure 2, 3, the emission of 1.0 µg/mL of acidic solution and SDS of RUP gave emission intensity around 400 FI, where the concentration of 1.2 µg/mL gives FI of 200, explain??

Reply:

There was a typing mistake in the title of figure 2. It was written that figures a,a' refer to 1.0 µg/mL RUP while the correct is 1.6 µg/mL RUP. This error has been corrected in the revised manuscript.

Comment 5: In the greenness study part, we noted that:

1. For the 1st procedure Green analytical procedure index (GAPI)

a) The author mentioned that there is NO extraction step (item NO 6), but what about adding 100 mL methanol, then sonicate for 30 min. (which required more energy exhaust)?? As well as in item NO 7, the original standard stock solutions were prepared in 100 mL methanol, I think it should be added to the scale of GAPI.

b) Part reagents and solvents: the amount of the used solvent, the stock solution prepared in 100 mL methanol for both drugs, moreover, and due its instability, MNT should be prepared every three days, in the same exhausted solvent. The wastes contain organic solvent as well as calculated each time to be involved in the GAPI scale too.

c) Instrumentation and waste's part: the actual volume of the wastes is calculated as the total repeated sample (as for example 10 mL each sample, containing NO of organic solvent, and reagent, multiplied in the whole repeated samples in the same ruing day), as in case of validation steps.

d) This should be considered as well as in the case of method II, table S2-B.

Reply:

a) NO.6, NO.7 and NO.8 has been removed as there is no extraction was done, no plasma extraction application. It is more like sample preparation but with simple procedures. For confirming this concept, we have contacted Prof. Justyna Płotka-Wasyłka (by email) who is the author that developed GAPI method, and she confirmed that the tablet extraction procedure here is considered a sample preparation step (c.f. plasma extraction).

NO.5 was edited in the revised manuscript as simple preparation (filtration) was done for the application on the dosage forms.

References:

- Nora A.Abdalah, Mona E.Fathy, Manar M. Tolba, Amina M. El-Brashy and Fawzia A. Ibrahim, (2021),green spectrofluorimetric assay of danterolene sodium via reduction method: application to content uniformity testing, R. Soc. Open Sci. 8:210562, doi: <http://doi.org/10.1098/rsos.210562>

- M.M. Tolba, M.M. Salim and Mohamed El- Awady, 2020), Simultaneous estimation of troxerutin and calcium dobesilate in presence of the carcinogenic hydroquinone using green spectrofluorimetric method. R.Soc. Open Sci. 8:201888, doi: <http://doi.org/10.1098/rsos.210188>

b) This part was calculated per sample that contained 1 mL methanol, so it's green shaded as the following reference indicated: [Analytical Eco-Scale for assessing the greenness of analytical procedures, trends in analytical chemistry, 37,2012, <http://dx.doi.org/10.1016/j.trac.2012.03.013>

c) Previous reports calculating the greenness for each sample in the section of (reagents and solvents) and (instrumentation and waste part). Only sample preparation is calculated for the whole process.

References:

- Agnieszka Gałuszka, Zdzisław M. Migaszewski, Piotr Konieczka, Jacek Namieśnik, Analytical Eco-Scale for assessing the greenness of analytical procedures, trends in analytical chemistry, 37,2012, doi: <http://dx.doi.org/10.1016/j.trac.2012.03.013>

- Nora A. Abdalah, Mona E. Fathy, Manar M. Tolba, Amina M. El-Brashy and Fawzia A. Ibrahim, Green spectrofluorimetric assay of danterolene sodium via reduction method: application to content uniformity testing, R. Soc. Open Sci., 2021, 8:210562. doi: <http://doi.org/10.1098/rsos.210562>

- M.M. Tolba, M.M. Salim and Mohamed El- Awady, Simultaneous estimation of troxerutin and calcium dobesilate in presence of the carcinogenic hydroquinone using green spectrofluorimetric method. R.Soc. Open Sci., 2020, 8:201888, <http://doi.org/10.1098/rsos.210188>

- Mona E. El Sharkasy, Rasha Aboshabana, F.Belal,M. wlash, Manar M.Tolba, journal of spectrichemica Acta part A:Molecular and biomolecular spectroscopy, 2022, 264, 120235, <http://doi.org/10.1016/j.saa.2021.120235>

d) The revised parts are also considered in Table S2-B.

Comment 6:

2. In the calculation of the Analytical Eco scale score

a) Methanol usage should be calculated for more than 100 mL, not only 1 mL.

Reply:

Methanol was added up to 1 mL in each sample to uniform the volume of methanol (i.e. to keep it constant in all flasks) and to reach the calibration range (for example, transferring 0.2 mL from the drug stock in one of calibration flasks, in another flask 0.4 mL , 0.6 mL,... till the last is 1 mL, by this there is no uniformity in methanol volume in all calibration flasks which add another factor to the calibration, so in each flask we complete to 1 mL by methanol (for example, 0.2 mL of drug + 0.8 mL of methanol, in another on adding 0.4 mL of drug + 0.6 mL of methanol so keeping the same volume of methanol to diminish the analytical error.

Comment 7:

3. In the last metric method related to NEMI based on evaluation of TRI and/or RCRA lists:

a) A 100 mL methanol, 1 mL of 0.1 M sulfuric acid (corrosive, and contained in the waste), that will affect, and they present in the TRI list as well as in the RCRA, and unfortunately, they will affect the quadrant of hazards as well, so the greenness profile using NEM index, for both methods should be revised to the real values.

Reply:

- Regarding sulfuric acid, 0.1M sulfuric acid is considered a corrosive material according to its safety data sheet so it is not green shaded in the corrosive quadrant in NEMI scale. Regarding the hazard, NEMI pictogram was modified according to the reviewer's comments.

- Methanol is used in a small quantity (1 mL) compared to other analytical techniques and it is considered green.

- The following scheme shows the greenness of solvents commonly used in analytical chemistry [M. Tobiszewski, Metrics for green analytical chemistry, Analytical Methods, 2016, 8, 2993-2999].

Comment 8: Finally, it is better to add further discussion to clarify the obtained results. Other miswriting and suggestions are present throughout the body of the pdf, kindly refer and reply.

Reply:

As recommended by the reviewer, further discussion has been added to the revised manuscript. In addition, the manuscript has been revised for miswriting and typos.

REVIEWER 3

Comment 1: English language should be substantially improved. There are many syntax and grammatical errors, mixing of tenses, etc. The text is difficult to understand at certain points. The manuscript is recommended to be read and corrected by a native English speaking scientist. Also, avoid capitalizing the chemical names.

Reply:

The manuscript has been thoroughly revised following the reviewer's recommendation.

Comment 2: Units: standard abbreviation for "hour" is h. There should be a space between number and unit (including M). Please be careful not to allow for a number and its unit in two subsequent rows. Use the nonbreaking space.

Reply:

As recommended by the reviewer, the manuscript has been revised considering the mentioned comments.

Comment 3: Title: perhaps the term "nanogram scale" is a slight exaggeration. LOD of methods were at the nanogram level, but not the linear range.

Reply:

We prefer to keep the term "nanogram scale" at the title because the values of LOD and LOQ are in nanograms and the start concentration of the linear range, which is 0.1 or 0.2 µg/mL, can be expressed as 100 or 200 ng/mL.

Comment 4: Introduction, penultimate paragraph: I found two papers on HPLC methods for simultaneous determination of rupatadine and montelukast in pharmaceutical formulations. Authors cited only one, the other is missing:

Redasani, VK, Kothawade, AR, Surana, SJ, J. Anal. Chem. 2014, 69, 384-389

DOI10.1134/S1061934814040121

Reply:

The recommended reference has been added to the revised manuscript (reference no. 27).

Comment 5: Experimental, subchapters 2.1 and 2.2: it is very unusual to give materials and equipment as bullet points. Please check the prescribed form of the journal in the Instructions for authors. Have you really used distilled water or perhaps deionized water? Mode of preparation should be given. For all chemicals, instruments and other equipment, the producer should be given. It is now missing in several instances.

Reply:

The “Experimental” section has been rewritten considering the recommendation of the reviewer. Both methods were carried out by double-distilled water not deionized water. This point has been clarified in the manuscript.

The names of the producers have been added to all chemicals, instruments, and other equipment in the revised manuscript.

Comment 6: Experimental, subchapter 2.4.1: the actual concentrations of analytes in standard solutions for calibration should be given for both methods. Line 47/48: what is meant by “plotted against the I drug concentration”? What is “I drug”?

Reply:

As recommended by the reviewer, the actual concentrations of analytes in standard solutions used for calibration has been added to the revised manuscript.

The letter “I” in line 47/48 was written by mistake. It has been deleted in the revised manuscript.

Comment 7: For simultaneous determination of both analytes, authors prepared only mixture 1:1. What about other ratios? Would it be possible to use Method II to determine both analytes at extreme ratios, i.e. if one of them was in great excess? Have authors checked that?

Reply:

This ratio was chosen because it is the ratio of the studied drugs in the commercially available pharmaceutical preparations. As recommended by the reviewer, other ratios have been checked and the obtained results are presented in Table S1.

Comment 8: Chapter 3.1: some results of methods optimization in the form of tables or graphs should be given

Reply:

As recommended by the reviewer, additional figures (Fig. 5 and Fig. 6) have been added to the revised manuscript to illustrate the results of the method optimization.

Comment 9: Validation of the developed methods & Table 1: LOQ is normally the lowest point of the calibration range, but in both methods they are much lower than the lowest point. Therefore, authors should further explain this discrepancy. How is the “percentage relative error” in Table 1 calculated?

Reply:

- According to ICH Harmonized Tripartite Guidelines, Validation of Analytical Procedures: Text and Methodology Q2(R1) (<https://www.ich.org/page/quality-guidelines>), LOQ is defined as: “The lowest amount of analyte in a sample which can be quantitatively determined with suitable precision and accuracy” and can be calculated as follows:

$$LOQ = 10 \sigma / S$$

where S = the slope of the calibration curve and σ can be calculated by a variety of ways, such as calculation based on the standard deviation of the y-intercept of the regression line or calculation based on the standard deviation of blank. In the current study, LOQ was calculated based on the standard deviation of the y-intercept of the regression line.

Following the recommendation of the reviewer, the values of LOQ and LOD have been recalculated using the second approach based on the standard deviation of blank and the results obtained are more acceptable than that of the first method. The new values have been added to Table 1.

- The % relative error was calculated by dividing the relative standard deviation over the square root of number of samples and then multiplying by 100. This explanation has been added to Section 3.2.

Comment 10: Table 2 and the corresponding text on page 10: different references ([6] or [7]) are given in text and in Table 2 for the comparison method used to determine accuracy. In any case, the comparison method should be briefly described in the Experimental part. What is meant by “Error” in Table 2 as there are no results for this entry?

Reply:

- The reference numbering has been corrected in the revised manuscript.
- Brief description of the comparison method has been added to the experimental part as recommended by the reviewer (Section 2.4.4).
- “Error” is the percentage relative error which is calculated by dividing the relative standard deviation over the square root of number of samples and then multiplying by 100. This explanation has been added to Section 3.2.

Comment 11: Table 3 and the corresponding text on page 10: What is meant by “% Error” in Table 3 and how was it calculated? There’s no explanation in the text.

Reply:

“%Error” is the percentage relative error which is calculated by dividing the relative standard deviation over the square root of number of samples and then multiplying by 100. This explanation has been added to Section 3.2.

Comment 12: Robustness testing (page 10): where are the results of testing? Some are given in the next paragraph, but most are missing.

Reply:

The results of the robustness testing have been presented in table 6 in the revised manuscript.

Comment 13: Selectivity testing (page 10): no results are shown, Moreover, there are many more excipients that could be used in the preparation of pharmaceutical formulations. Also, it would be good to see if the quantification of both analytes is possible at ratios different from 1:1

Reply:

- The selectivity of the proposed method has been proven by the high values of the percentage recovery (%found) for the analysis of prepared and commercial tablets (Tables 4 and 5) which indicate the absence of interference from tablet excipients.
- The excipients used were selected as representative examples of common tablet excipients. Lactose and maize starch are both examples of fillers and binders, talc powder is an example of glidants, and magnesium stearate is an example of lubricants. There is no specific reference for a fixed type and ratio of tablet excipients as that depends on several factors such as the dose of active ingredients, total tablet weight, tablet disintegration, adsorption effects etc. Moreover, most of the pharmaceutical companies does not mention the nature and amount of tablet excipients in the internal leaflet of their tablet preparations. Therefore, these components were selected just to study any possible interference of common tablet excipients to prove the selectivity of the developed method.
- The ratio 1:1 was chosen because it is the ratio of the studied drugs in the commercially available pharmaceutical preparations. As recommended by the reviewer, other ratios have been checked and the obtained results are presented in Table S1.

Comment 14: Tables 4 and 5 and the corresponding text on page 11: again, discrepancy in references for comparison method in the text and in Tables. What is meant by “% Error” and how was it calculated?

Reply:

- The reference numbering has been corrected in the revised manuscript.
- “%Error” is the percentage relative error which is calculated by dividing the relative standard deviation over the square root of number of samples and then multiplying by 100. This explanation has been added to Section 3.2.

Comment 15: Table 6: at GAPI pictograms, Table 7 is mentioned which is missing. What is meant by “hermitization” – probably hermetization?

Reply:

- The table numbering has been corrected in the revised manuscript. GAPI pictograms are presented in Table 7 while GAPI parameters for the developed methods are presented in Table S2.
- Hermitization is related to the evaporation of solvents and how it could be toxic to the environment. [Van Aken Koen, Strekowski Lucjan and P. Luc, EcoScale, a semi-quantitative tool to select an organic preparation based on economical and ecological parameters, J Beilstein Journal of Organic Chemistry 2 (2006) 3] and [Kanakan Parvathi Kannaiyah, Abimanyu Sugumaran, Hemanth Kumar Chanduluru, Seetharaman Rathinam, Environmental impact of greenness assessment tools in liquid chromatography – A review, Microchemical Journal 170 (2021) 106685].

Comment 16: Finally, I would suggest that the results of developed methods should be compared to results of a rather different method (HPLC) that is considered more selective than derivative spectrometry used by authors for comparison.

Reply:

As recommended by the reviewer, the comparison method has been changed to another more selective HPLC method [A. Jani, J. Jasoliya and D. Vansjalia, Method development and validation of stability indicating RP-HPLC for simultaneous estimation of rupatadine fumarate and montelukast sodium in combined tablet dosage form, J Int Pharm Pharm Sci, 2014, 6, 229-233]. The corresponding results have been updated in Tables 2, 4 and 5. In addition, the statistical comparison of the results including Student *t*-test and Variance ratio *F*-test have been recalculated.